# The Disputable Costs of Sleeping

**DOI:** 10.3390/biology14040352

**Published:** 2025-03-28

**Authors:** Mourad Akaarir, M. Cristina Nicolau, Francesca Cañellas, Jose A. Rubiño, Pere Barceló, Antonio Gamundí, Aida Martin-Reina, Rubén V. Rial

**Affiliations:** 1Balearic Islands Health Research Institute (IUNICS), Universitat de les Illes Balears, 07122 Palma, Spain; mourad.akaarir@uib.es (M.A.); cristina.nicolau@uib.es (M.C.N.); pbarcelocaldentey@gmail.com (P.B.); antoni.gamundi@uib.es (A.G.); aida.martin.reina@gmail.com (A.M.-R.); 2Balearic Islands Health Research Institute (IUNICS), Hospital Universitario Son Espases, Universitat de les Illes Balears, 07122 Palma, Spain; francesca.canellas@ssib.es (F.C.);

**Keywords:** sleep costs, vital needs, optimal prey, sleepiness

## Abstract

Many authors have stated that sleep detracts time used for foraging, defense, and anti-predatory activities. Therefore, sleep must provide some compensating advantage. Instead, we show that the sleep-related reductions in food intake and reproductive activities may be in fact benefits, both for the individual and the species. Furthermore, we show that the optimal prey are the immature, weak, sick, and senescent animals and rarely the sleeping individuals. Indeed, the reduced amounts of sleeping time observed in prey animals occurs not because of an antipredation evolutionary pressure, but mostly because of the need for time to eat and digest the high-cellulose contents of the herbivores’ diet, a set of tasks that leaves reduced time to sleep. In summary, no animal restrains their vital activities for sleeping, and this means that the need for sleep is low on the list of the vital activities. In fact, sleeping basically consists of doing nothing, and no live being can die from insomnia. Instead, what is important is maintaining efficient wakefulness, which can be achieved only after a sufficient amount of sleep.

## 1. Sleep in Animals

In behavioral terms, sleep is currently defined as a state showing eight differential traits: (1) quiescence, (2) reversibility, (3) specific sleeping places, (4) specific body positions, (5) circadian organization, (6) homeostatic regulation, and (7) being a pleasing state [1,2,3,4,5]. The raised sensory thresholds are also included, but the present review will show that this trait is not as important as currently believed. On the other hand, most behavioral traits of sleep can be observed in wakeful resting animals. Of course, the electrophysiological methods allow distinguishing between sleep and wakeful rest but, although these methods have been applied to invertebrates—mostly to drosophila flies—no electrophysiological method can tell whether an immobile fly is asleep or is merely resting [6,7].

But, most important is that a complete definition of behavioral states in all animals cannot be attained without making reference to its antagonist state: wakefulness.

Wakefulness has been defined as being in a state with the capacity to perceive environmental stimuli and respond them with adaptive responses [8]. However, two kinds of wakefulness exist: unconscious and conscious. So, the definition must be completed. For instance, a photocell can perceive the presence of a light beam and then it opens a door as a response. But, this activity does not signify that the photocell is awake and in no way can be equated to the complete and true wakefulness as observed in some—but not all—animals. The true difference between the wakefulness of a photocell and that of a human subject lies in the obvious unconsciousness of the former and the consciousness of the latter. But, we may compare two activities: the human myotatic reflex and the positive response of an infant after seeing a sweet. Of course, both responses are perceived and adaptive. But, while the response to sweets is conscious, no true difference exists between a photocell opening a door and the myotatic reflex. Both are unconscious responses.

The definition of consciousness has been addressed in many studies following empirical procedures. From such studies, the presence of conscious wakefulness has been recognized in animals showing play, detour and anticipatory behaviors, learned taste aversion, emotional fever, and tachycardia [9]. In general, consciousness is the capacity of individuals to make some kind of space/time trip and process the result(s) of future actions. It has been found that mammals, together with some birds (Corvidae and Psittacidae, for example), can show conscious adaptive responses [10]. Likewise, some hints of consciousness have been observed in reptiles [11] but not in amphibians, but, of course, are evident in mammals. In summary, after analyzing the presence or absence of the described capabilities in different animal groups, it was concluded that only mammals, birds, and reptiles show signs of consciousness and true sleep, with the full set of traits defined in the previous paragraph [12]. The distribution of consciousness in the animal kingdom is shown in Box 1.

Box 1Behavioral states in animals.Homeotherms: Mammals and birds: unconscious (true) sleep, conscious (true) wakingPoikilotherms: Reptiles: Conscious wakefulness no true sleep, cool restAmphibians: Unconscious cool rest, unconscious wakingFish: Unconscious rest, unconscious wakingInvertebrates: Unconscious rest, unconscious waking

In summary, the presence or absence of consciousness is the basic difference between the states of sleep and wakefulness in poikilothermic (cool-blooded) and homeothermic vertebrates (warm-blooded; mammals and birds): both can perceive and react to environmental stimuli with adaptive responses, but only homeothermic vertebrates and reptiles show conscious responses. On the other hand, poikilothermic animals show two different states: unconscious sleep-like and unconscious wake-like states. Reptiles are exceptional: they are true poikilotherms but can be facultative homeotherms. So, they can show conscious wakefulness. Instead, homeothermic mammals and birds show unconscious sleep and conscious wakefulness, i.e., true sleep and true wakefulness. In other words: some sub-mammalian vertebrates can show the traits defining the mammalian sleep behavior, but they show neither true unconscious sleep nor true wakeful consciousness. Nevertheless, the analysis of the behavioral states of birds is a complex question that merits a detailed analysis and, leaving apart some particular cases, is not addressed in the present review.

Drowsiness is another behavioral state. It is currently described as a definite and stable state of wakefulness that is opposed to alert wakefulness. In all ruminants, for example, the arousal threshold to audio stimulation remained low during rumination but increased by a factor of approximately ten during slow-wave sleep. Therefore, rumination is associated with wakefulness and drowsiness. Rumination may continue even during slow-wave sleep (SWS) as defined by the ECoG pattern. In this case, however, the ruminating rate is slowed but can also be frequently observed when a ruminant lays actively awake but slightly drowsy [13].

In summary, five behavioral states have been observed in animals: (1) rest, (2) drowsiness, (3) slow-wave sleep, (4) REM sleep, and (5) wakefulness. Additionally, the last one, wakefulness, can be subdivided into 5.1. unconscious wakefulness and 5.2. conscious wakefulness. We see that the definition of behavioral states is complex. The described problems with behavioral state definition are observed not only in invertebrates [9]; doubts have also been raised regarding the existence of true sleep in poikilothermic vertebrates [8,14,15,16]. In their natural environment, reptiles seem to be asleep during night-time. But, their true state is uncertain. In fact, the presumed reptilian sleep is a passive state of “voluntary hypothermia”, with poor motor coordination, reduced sensory sensitivity, and incapacity for rapid reversibility [12,14,15,17]. However, the reversibility of sleep is strictly dependent on the body temperature of reptiles, and, in fact, the reversibility of the mammalian sleep has been defined only for distinguishing sleep from coma [18] and other incapacitated states. Despite these details, the existence of sleep, and even the existence of NREM and REM, has been affirmed in reptiles [19,20,21]. However, these studies disregarded the consequences of the passive nocturnal hypothermia typical of poikilothermic animals, in which the presumed sleep is a passive state of dormancy as a result of reductions in body temperature and physiological activity. So, the presumed reptilian sleep is a state different from the active sleep of homeothermic mammals [22,23,24]. Furthermore, the efforts to find the two states, NREM and REM, in nonmammals produced rather poor results [19,20,21], widening, therefore, the gap between mammalian sleep and the presumed sleep (the dormancy) of poikilothermic vertebrates.

## 2. The Birth of Mammalian Sleep

Many authors recognize that mammals evolved because of the so-called evolutionary nocturnal bottleneck (NEB) [25]. At the boundary of the Cretaceous–Paleogene, ~250 million years ago, some small diurnal reptiles developed incipient metabolic endothermy. This allowed them to extend their activity first to crepuscular hours, and, after further metabolic improvements, they ended up as full homeothermic mammals, capable of being active during the night-time. Furthermore, to improve their visual sensitivity in the dark, they abandoned the visual filters that protected the eyes of their reptilian ancestors from the most energetic and dangerous fraction of diurnal light. However, such an abandon increased the risk of blindness in the case of casual exposure to daylight. So, primitive mammals had to be strictly nocturnal, as most mammals are today. At present, the high visual sensitivity in scotopic environments and the development of endothermy are considered key factors in the evolution of mammals [26,27,28].

Furthermore, early mammals were forced to share their environment with terrible predators, the dinosaurs. So, the dangers of blindness and predation exerted a high pressure not only to restrict their activity to dark time but also to remain immobile in lightproof burrows during times of light. Only after the Cretaceous–Paleogene extinction event, about ~66 million years ago, did most dinosaurs become extinct, and the diurnal niche was free of competitors. This facilitated the return to diurnal activity in some mammalian species. They simply needed to change their chronotype again and recover the visual filters for supporting their exposure to diurnal light. To summarize, we would like to ask to the reader what we should call a paralyzed state with closed eyes that was repeated, day by day, during uncountable millions of years? Undoubtably, the answer to the question is sleep [8]. On the other hand, birds also evolved from reptiles—they are often considered to be feathered reptiles. But, neither their homeothermy nor their sleep are related to mammalian homeothermy and sleep. Therefore, the present review only deals with mammals and, only after considering that birds also show the eight traits of behavioral sleep, we refer to some examples of avian sleep. Indeed, the behavioral traits of avian sleep are identical to those of mammalian sleep.

An important theoretical consequence of the described facts is that, far from currently believed, sleep—and similar sleep-like states—is polyphyletic, i.e., it has appeared four times (at least in invertebrates, sub-mammalian vertebrates, modern mammals, and, independently, in birds, in which no nocturnal bottleneck has ever been reported). These facts might signify that if we take human sleep as a norm, true sleep only exists in homeothermic animals, mammals, and birds. Therefore, we may discard the sleep-like state of nonmammals.

Because of these facts, the present report deals only with the presumed costs of mammalian sleep, and we attend only to sleep as a behavior. This means that our study combines NREM and REM in a single state: sleep. This is reasonable: First, the words “NREM” and “REM” are always appended to the word “sleep”, meaning that, in behavioral terms, sleep is a single state. Along the same vein, we may remember that the widely recognized two processes of sleep regulation only refer to sleep with no distinction between substates [4]. Furthermore, the existence of NREM and REM remained invisible to humans during thousands of years: they were considered simple sleep, without additional considerations.

## 3. Causal Relationships in Sleep

In mammals, the decision to be awake or asleep depends on several internal and environmental factors. Among the first factors, the circadian organization and the homeostatic processes are essential [5,6]. But, sleep also depends on foraging, predation, reproduction, and many other environmental factors.

However, we have observed that when describing the relationship of sleep with other behavioral activities, many reports do not distinguish between causes and consequences. Please consider the following statements:

I. Regarding sleep and foraging:

I.(a) Spending time sleeping may cause reductions in foraging time. Reduced food intake would be the consequence.

I.(b) Hunger causes increased exploration for food, with consequent reductions in sleeping time. Sleep reductions would be the consequence.

II. Regarding sleep and reproduction:

II.(a) The time spent sleeping causes reductions in the time devoted to reproductive activity. Such reductions would be the consequence.

II.(b) The time spent in reproductive activities causes reductions in sleeping time. Sleeping loss would be the consequence.

III. Regarding sleep and predation:

III.(a) The sleep-related rise in sensory thresholds causes increases in predation risk. Such increases are the consequence.

III.(b) Predatory stress causes reductions in sleeping time. Such reductions are the consequence.

We see an evident contradiction within each one of the three pairs of statements. We ordered the questions putting, for each pair, (A) sleep as a presumed cause (sentences I.(a), II.(a), III.(a)) and (B) some factors with capability to cause modifications of sleep (sentences I.(b), II.(b), III.(b)). The three groups of sentences show that the behavioral states of animals are in fact flexible and suggest the convenience of always considering the factors that can modify the causes and consequences of the behavioral states.

## 4. Sleep and Vital Activities

A huge number of reports confirm that sleep causes reductions in the time devoted to foraging, procreation, and anti-predatory activities, i.e., agreeing with sentences I.(a), II.(a) and III.(a) [29,30,31,32,33,34,35,36,37,38,39,40,41,42]. Please note that the cited list of references is only a sample; the total number of reports claiming that sleep causes diverse physiological–psychological limitations is enormous. However, and to the best of our knowledge, no report has explained how such a hypothesis was supported: it seems that the sleep-related reductions in vital activities were taken for granted: obviously, sleep is incompatible with foraging, defense, reproduction, and related activities. Altogether, these reports consider that (1) sleep is a handicapped state; (2) sleep must provide important advantages to compensate for the presumed handicaps (otherwise, it would have been removed by natural selection); (3) sleep seems to rank first on the list of biological needs, and the remaining activities only play subsidiary roles (otherwise, sleep would have no power to interfere with them); (4) foraging, reproduction, defense, and related activities, together with sleep, completely fill the total daily time. Therefore, increasing the time devoted to sleep implies reductions in the time available for the remaining activities. The present review disputes these four statements.

Of course, a few reports have defended an inverse relationship: that many environmental or internal factors cause disturbances in the expression of sleep. For instance, hunger causes increased exploration, therefore reducing the total sleeping time [41,42,43] agreeing with sentence 1b. It should be noted, however, that the number of reports supporting these inverse correlations is quite low when compared with the huge number of reports supporting a causal role of sleep in the competition with foraging, reproductive, and anti-predatory activities. These reports support the existence of activities outside of the main triad (foraging, defense, and reproduction). These activities, however, are of reduced importance when compared with those of the triad.

## 5. Sleep, the Metabolic Syndrome, and Excessive Foraging

The reductions in foraging and foraging-related activities are some of the frequently claimed hindrances of sleep. To the best of our knowledge, however, no report has considered the health disturbances caused by metabolic syndrome and its relationships with sleep and foraging. Metabolic syndrome involves a constellation of pathological signs resulting from the disruption of the circadian rhythm [44,45,46,47], and from excessive food intake [48,49,50,51,52,53,54]. People suffering metabolic syndrome show high morbidity and mortality [55,56,57,58] because of obesity, insulin resistance, constipation, hypercholesterolemia, hypertension, heart attacks, strokes, etc. These problems are currently observed in stressed people, in shift workers, and in trans-meridian travelers, in other words, in people suffering sleep disturbances. Therefore, if such disturbances contribute to the reduction in foraging time and foraging-related activities, the consequence is a mitigation of metabolic syndrome, i.e., an advantage and not a hindrance. Moreover, since the pioneering work of McKay et al. [59], it has been well recognized that caloric dietary restriction improves the general health and extends the lifespan of many species, including several rat and mouse strains, hamsters, as well as nonhuman and human primates [60,61,62,63,64], including sub-mammalian species [65] and even invertebrates [66]. Likewise, many studies have demonstrated that reducing the body weight of experimental animals to about a 70% of the normal “ad libitum” feeding (undernutrition without malnutrition) delays the progression of a variety of age-related diseases in nonhuman primates [67,68,69,70,71,72] and helps with maintaining a youthful state up to advanced ages [73,74]. This conclusion does not mean to sleep more and to eat less; as already said, all considerations reflected in this paragraph are relative: the reductions in foraging must always lead to healthy undernutrition and never to pathological malnutrition.

Altogether, these findings show that reasonable reductions in food intake constitute a solid counterproof dismantling the presumed problems with sleep-related reductions in foraging time. Indeed, if sleep curtails foraging and foraging-related activities, the overall consequences would be positive, and no compensatory advantage should be demanded to sleep. But importantly, the described facts show that excessive food intake, together with circadian disturbances, are the main factors causing metabolic syndrome. Therefore, sleep cannot be a causation of the deleterious consequences of MS. They are, instead, consequences of a disordered control of foraging, a problem that is frequently correlated with sleep reductions [75,76,77,78], i.e., the opposite of the general claim. This means that, on the list of life-sustaining activities, the eventual advantages provided by sleep, rank lower than those of foraging.

## 6. Sleep and Reproductive Efficiency

Many authors have found that sleep detracts time from reproductive activities. Therefore, we should describe a case of some mammals in which sleep impairs their reproductive activity. However, we begin by discussing an interesting example observed in birds. The following lines describe the relationships between sleep and reproductive activities in the pectoral sandpiper *Calidris melanotos*. We selected a bird as a paradigmatic example of the competence between sleep and reproductive activity because of the wide diffusion of the Lesku et al. report [79] (259 cites). These authors found dramatic reductions in sleeping time during the reproductive season in the male pectoral sandpiper. In addition, they found that the number of sired chicks inversely correlated to the total sleeping time of the males. Therefore, it seems that sleep truly interferes with the reproductive activities of some birds. So, reductions in total sleeping time increase reproductive success. It should be noted, however, that the male pectoral sandpiper is polygamic, which means that the female, after having been fertilized, is abandoned. Then, the male searches for another female with which to breed, so that the male can generate an additional clutch.

However, one may ask, what about the reproductive success of the female? Obviously, it must be proportional to the number of eggs in the clutch. It is evident, however, that because of polygamy, the reproductive success of the females is independent of the success of the males. It is also evident that the female must carry on with the activities that determine her reproductive success: oviposition, incubation, alimentation, defense of the newborn chicks, etc. So, no relation exists between the reproductive success of males and that of females. It only depends on the number of eggs posited by the female.

Nevertheless, it is well known that reproductive success cannot be improved by increasing the reproductive efforts of the individuals. Moreover, in 1944, Moreau [80] observed that ‘it is far from certain that the bigger clutch is always more to the good of the species’ and that ‘a greater abundance of young may induce a disproportionately, greater attention from predators’. Moreover, ‘in a climate that is uncertain, the effects of a bad season might be more disastrous on bigger broods that were adapted in size to the supply of the best seasons and that the amount of food brought to the nest does not increase in proportion to the number of young’. Likewise, Lack [81], observed that ‘the survival probability decreases with increasing the litter size, because the amount of food parents can provision to their offspring is limited’.

So, a high number of sired broods by males is of reduced importance; high reproductive success of a female might increase the predatory risk and the survival of the descendants in bad times. Therefore, the reproductive success of males showing reduced amounts of sleep may even reduce the success of the species: too many clutches lead to a higher predation risk and to a risk of reduced survival during bad seasons.

In summary, the total sleeping time of the male may widely vary and may not increase the success of the species. Therefore, sleep seems to rank below reproduction on the list of vital needs. And, as Moreau and Lack observed, curtailing the reproductive efforts of males may increase the success of the species. Indeed, a small clutch may remain unnoticed by predators, and in a clutch with a low number of newborns, survival may be better during bad seasons. Therefore, we can summarize the above as follows:

Reduced total male sleeping time → High number of sired chicks

High number of sired chicks → higher predation risk of the nest

Higher predation risk of the nest → Reductions in clutch size

Reduced clutch size → reduced predation

Reduced predation → higher survival

Total: high amounts of total sleeping time → reduced predation and increased survival in bad seasons. So, if sleep detracts from the time used for reproductive activities, the net result is an advantage, and no reason exists for compensating the increase in total sleeping time.

The described study was performed on a bird, but multiple reports show identical results and consequences in mammals [82,83,84,85,86,87,88,89,90], Of course, the mammalian female must carry offspring alone through the efforts to support the development of the internal embryo, parturition, lactation, and the defense of the brood. So, as we observed in the Pectoral Sandpiper, the reproductive success of mammals depends on the reproductive efficiency of the female and not on the reproductive success of the male. This is well known in humans, in which the mother is often alone in performing the tasks required for rearing her infants.

Nevertheless, one may ask about the relation between sleep and reproductive success in mammals. We analyze this question in the case of humans. Obviously, an immense proportion of humans have been conceived during the pre-sleeping period. The total sleeping time of most humans depends on alarm clocks; thus, one must conclude that the success of human reproductive activity implies reductions in total sleeping time. Thus, we can assert that the immense reproductive success—the overpopulation—of mammalian species is favored by reductions in total sleeping time. To conclude, the presumed hindrances of sleep in both pectoral sandpipers and humans—and probably in all mammals and birds—do not exist. And no reason exists to claim compensations is required for the presumed costs of sleeping.

## 7. Sleep and Predation

It is currently affirmed that sleep increases predation risk [91,92,93,94,95,96,97]. Such a high number of authors is indicative of a general belief that. sleep is a defenseless state. Nevertheless, some authors also considered that the immobility of sleep reduces the risk of being detected by predators [98,99], a possibility that has been considered by some of the authors that, at the same time, defend sleep-related increases in predation risk. See Lima and Rattenborg [96] as an example. As we previously noted when commenting the relations between foraging and sleep, sleep-related immobility is another example of the flexibility in the definition of the causes and consequences controlling the expression of behavioral sleep.

Two facts support the presumed increase in sleep-related predatory risk: (1) the sleep-associated rise in sensory thresholds that increases the success of predatory attacks and (2) the reduced amounts of sleep observed in prey, reductions that can be interpreted as the result of an evolutionary pressure minimizing the risk of predation. Please note, however, that these two facts only indicate that sleep is suspected to increase the predatory risk. As far as we know, no report has provided an analysis of successful attacks on sleeping prey and contrasted it with similar attacks on wakeful prey. So, the sleep–predation relation may be a reasonable guess but, for the time being, lacks experimental evidence. Regarding the inverse correlation between predation and total sleeping time, the next paragraphs provide a detailed analysis to test whether such a correlation really exists.

### 7.1. Predators and Prey: Costs and Benefits

It is often stated that in the competition between predators and prey, the cost for prey is death, while the benefit is survival. In contrast, predators only risk a meal. Therefore, the costs and benefits of prey and predators seem to be extremely asymmetric. Undoubtedly, such a belief is wrong: predators kill prey with claws and fangs, but prey kill predators through hunger, starvation, and exhaustion.

Lotka and Volterra [100,101] observed that the populations of predators and prey are dependent on each other, and both populations show continuous cyclic oscillations. However, in stabilized environments, the population sizes remain approximately constant. If predators improve their killing efficiency, the number of prey certainly decreases. However, such a reduction leads to food scarcity and hunger in the population of predators, which consequently reduces. But, after a lag, the reduced number of predators allows the survival of more prey, which successively promotes increases in the predators’ population. The cycle is endlessly repeated. Therefore, the costs and benefits of predators and prey are balanced (Figure 1).

Of course, the original equations of Lotka and Volterra were formulated for a single predator–prey species pair, and the complexity of the relationship increases with the number of species co-predating or being co-preyed [104,105,106,107]. But the principle holds; apart from the exceptional cases of extinction, the population and the costs of predators and prey are stable. Therefore, we deny the popular belief: given that the average population of predators and prey remains unaltered, the costs must be equivalent.

### 7.2. The Concepts of Optimal Foraging and Optimal Prey

The concept of optimal foraging measures the profitability of a prey in terms of the energetic gains per unit of the predators’ foraging efforts. However, while most reports analyzed the optimization of the predator’s energetic expenditures [108,109,110], only a small number of reports have analyzed the prey expenditures. But most important: no report has been published—to the best of our knowledge—including sleeping animals within the group of optimal prey. Instead, a high number of reports recognized that the optimal prey includes the immature, weak, sick, and old individuals, but not the fittest adults [111,112,113,114,115,116,117,118]. It may be argued that predators arbitrarily choose prey without considering their health or their fitness. But, this is wrong: from the prey side, it is bad enough to be killed by a predator, but it is worse if that prey is an adult with full defensive and reproductive capacities, as its eventual descendants will also disappear. On the contrary, pruning the immature, sick, or old individuals, all with low reproductive efficiency, would have few consequences for the population. Furthermore, attacking a full-grown fit prey is problematic: such prey demands high hunting efforts and the probability of the predator being hurt is increased because of the defensive tools that fit prey use in the course of the attack. Therefore, the interests of predators and prey coincide. From these facts, it is believed that a sophisticated communication procedure has appeared for mutual benefits, a procedure defined as the handicap principle [119]. It takes the form of honest and costly signals emitted by fit wakeful prey informing the predators on the convenience of abandoning unprofitable and risky attacks. Note that the honesty and the cost of the signals produced by prey are immediately recognized by predators as indicative of prey fitness and the profitability of the hunting effort. Of course, less-fit prey might emit false signs of fitness, but the predator easily recognizes the fake. A well-recognized example of such interactions is the stotting behavior performed by many herbivores when escaping from predators. Such behavior is, in fact, an unnecessary and risky ostentation of fitness that only can be explained as a message sent to predators: “I’m too fit for you and I’m going to dearly sell my life”. Any predator observing such a display wisely turns to attack another not-so-fit prey [120]. Notably, the report defining the handicap principle has received, at present, 3257 citations, which is indicative of wide acceptance among scholars. Of course, predators may try to attack any prey who comes along, independent of its fitness. This might be true in some cases in which the predators may be stalking unaware prey, but honest signs of fitness are evident in the first stages of the attack, and the predator should react wisely, abandoning a costly and risky attempt instead of pursuing more likely success. To conclude, we have described the signs marking the maximal benefit for predators: they do not blindly attack prey without considering the health and fitness of the prey; only with rare exceptions, they recognize the fitness of their eventual prey and select the optimal ones, that is, the immature, weak, sick, and old individuals.

### 7.3. Sleep and Predation: Causes and Consequences

The perception of any increase in predatory risk—and stress, in general—causes sleep restriction in prey. This means that sleep loss is the consequence of stressful circumstances. For example, no animal would dare to sleep under predatory siege. But, many authors claim the opposite, i.e., that sleeping causes increased predatory risk. So, we have two contending opinions: some sleep traits might cause increases in risk. Alternatively, a high predation risk might cause reductions in sleeping time.

In principle, both sides seem to be right. As well recognized, the stress and the perception of predatory danger causes sleep suppression in all animals [121,122,123,124]. Of course, such a perception may occur only in prey sleeping in dangerous places, for instance, in the open, and much less in burrowing species. On the other hand, any animal with the incapacity to perceive the risk must be an optimal prey. So, to study the issue, we will analyze first whether the sensory thresholds are truly raised during sleep.

#### 7.3.1. The Perception of Predatory Risk Using Contact Sensory Receptors

The sensory organs of live beings can be classified as contact receptors and tele-receptors. Many mammals sleep with conspecifics in a huddle, and a collective startle might be indicative of an immediate predatory attack. But, most likely, the startle determined by physical contact would arrive too late to be useful, and the same would occur in the case of a lone sleeper contacting a predator. Obviously, the predation would be unavoidable when it is perceived by contact receptors. Only tele-receptors can allow a sufficient delay to reduce the risk. Consequently, we make no further mention of contact receptors. Instead, we consider tele-receptors—vision, audition, and olfaction—and their importance in minimizing predation risk.

#### 7.3.2. Vision

Sleep-related eye closure is almost universal in animals possessing eyelids and, when it is complete, may cause a total blockade of the visual input. However, the visual block lacks consequences either for the survival of burrowing species or for those that sleep during dark time. Nevertheless, some mammals and birds are capable of so-called lagophthalmos, sleeping with open eyes or partially or unilaterally closed eyes [125,126,127,128].

It is believed that most cases of sleeping with open eyes allows the maintenance of partial vigilance. So, burrowing animals, the ones that sleep during night time, and those capable of lagophthalmos use such traits as defensive mechanisms that may counterweigh the sleep-related risk of predation. So, the risk of a sleep-related complete eye closure must be high only in solitary animals sleeping in the open. But such animals are extremely rare and may only occur when the individuals accumulate a sleep debt whose consequences will be analyzed in next paragraphs. Indeed, most animals that sleep in the open are grouped in herds in which collective vigilance also counterweighs the consequences of sleep-related eye closure [129,130]. For example, individuals placed on the edge of the herd show increased visual vigilance during light time but rely mostly on audition during night [131,132].

To summarize, visual and acoustic vigilance may be an antipredation measure for herds but is much less so for the low number of species that, being incapable of lagophthalmos, also sleep in the open during light.

#### 7.3.3. Audition

It has been affirmed that the mammalian auditory system is continuously on duty [131,132]. The mismatch negativity potential (MMN) is an auditory event-related potential that appears after sounds that differ in power and/or frequency from the background noise, and although the MMN shows significant reductions in amplitude in sleeping subjects, the potential can still be recorded [133,134,135]. Interestingly, the MMN amplitude remains high across states in sleeping newborns, i.e., in optimal prey [136,137,138,139,140,141,142,143]. Altogether, it seems that the MMN is a sign of auditory vigilance that appears in a population in which the single defensive activity consists of crying for maternal help. In addition, it has been affirmed that “no neuron belonging to any auditory pathway level or cortex was observed to stop firing during sleep…” [132]. Moreover, it has been found that during all stages of sleep, the mismatch negativity (MMN) in response to emotional syllables can always be detected. In the same way, it has been reported that “the sleeping brain is less and less considered as a passive and isolated resting organ” [143]. So, it seems that the MMN and its associated mechanisms continuously scan the environment, searching for odd sounds and deciding whether they are trivial or indicative of danger, possessing the capacity for interrupting the sleep and taking appropriate defensive measures [144,145]. Furthermore, Turker et al. [146] instructed several subjects to frown or smile depending on the stimulus type (happy vs. fearful). As a result, the subjects produced accurate behavioral responses in all sleep stages according to the emotional load of the stimuli, Moreover, the MMN evoked in response to emotional syllables can always be detected during all stages of sleep. To summarize, “the sleeping brain is less and less considered as a passive and isolated resting organ” [143]. Indeed, the behavioral and brain responses to verbal stimuli reveal transient periods of cognitive integration of the external world during sleep [145]. In sum, “no neuron belonging to any auditory pathway level or cortex was observed to stop firing during sleep” [132]. In the same way, it has been reported that “the sleeping brain is less and less considered as a passive and isolated resting organ” [147]. To summarize, it seems that the MMN is an efficient mechanism for responding to sleep-related predatory risk.

#### 7.3.4. Olfaction

Carnivores (predators) use glandular secretions, urine, and feces to mark their territory [148,149], and many prey animals avoid the places scented by carnivores [150,151,152,153,154,155]. Of course, the reactions of olfactory avoidance are evident in wakeful individuals, but many reports recognize that olfaction remains active during sleep in both humans (but see [156]) and animals [157,158,159]. The olfactory sensitivity of humans is rather reduced when compared with that of macro osmatic mammals. However, the olfactory-evoked local field potential (LFP) shows increased amplitude in sleeping humans [160,161,162,163]. Most interesting are the experimental consequences of odor stimulation in sleeping subjects. For instance, sleeping newborn humans and rats, i.e., optimal prey, respond to olfactory stimulation with sniffing movements and specific facial and autonomic responses [164,165]. So, it seems that the olfactory LFP works like the auditory MMN [166], and some reports state that olfactory stimulation in sleeping humans may cause behavioral awakening and micro-switch closure, as well as changes in heart rate, EMG, respiration, and EEG [167].

### 7.4. Sleep-Related Sensory Shutdown: Is It Real?

After having described the importance of tele-receptors in sleeping animals, we consider that the current belief regarding sleep-related helplessness is rather disputable. This is particularly evident for audition: the auditory system of sleeping individuals is always in standby mode, continuously monitoring the environment and deciding on the convenience of continuing sleeping or waking up [168]. We thus conclude that the activities of auditory and, with lesser importance, the olfactory systems, during sleep may be important defensive tools for prey, counterweighing the predators means of attack, and playing a significant role in the stability of the predator–prey relationship. Nevertheless, we already observed that sleep only is suspected to increase the predatory risk (Section 6), that sleeping animals only are suspected of being at high predatory risk, and no proof exists of them being optimal prey. As predators must select the easiest prey, they must select the immature, weak, sick, and old individuals. Scotophobia—the fear of darkness—and agoraphobia—the fear of open spaces—may provide interesting explanations. Scotophobia only appears as a transient pre-sleep phase in human infants [169], i.e., optimal prey. In contrast, agoraphobia—the fear of open spaces—has a prevalence of 1.4% in adult humans, which is high enough to be included as a separate disorder in the DSM IV catalog of anxiety disorders. The maximal incidence of agoraphobia occurs around 30–44 years of age, with a reduced incidence in those under eighteen [170], i.e., it only appears in the fittest wakeful adults. The difference between the two phobias clearly tells us that in humans, predation during dark time—sleeping time—is truly dangerous for infants, but less for wakeful full-grown adults which only fear open spaces where they

To summarize, the sleep-related predatory risk seems a reasonable guess but lacks confirmatory evidence: all animals search for comfortable and safe sleeping places where the predatory risk is low. On the contrary, the maximal risk appears in open spaces and in wakeful animals. Nevertheless, both the auditory MMN and the olfactive LFP—remember its high amplitude in infants—may be important aspects in sleeping mammals that contribute to minimizing the predatory risk.

### 7.5. Do Predators Sleep More than Prey?

The previous paragraphs showed the reduced importance of the presumed sleep-related rise in sensory thresholds. But we must also analyze whether evolutionary pressure truly reduces the total sleeping time in prey animals, as defended by many authors [30,91,92,93,94,95,171]. The following headings, we analyze the different factors contributing to the modification of total sleeping time.

### 7.6. Body Size and Predation Risk

A factor exerting a strong influence on the predator–prey relationship is body size. A simple reasoning explains that small animals can be predated by larger predators. Oppositely, the number of small predators with the capacity to attack bigger prey must necessarily be lower [118,172,173,174,175,176,177,178,179,180].

Several reports analyzed the body size ratio of predators and prey. They found that the geometric mean of the predator’s size increases with the size of the prey, an increase that is less than linearly proportional. Although predators tend to select smaller prey, smaller prey also occasionally dare to attack big predators. However,, when the difference begins to increase, the prey starts to leave the range of optimal prey [178,179,180,181]. This may be the case of adult elephants weighing over 1000 kg—megaherbivores—in which the in the Zoo setting—with presumable low predatory risk (but also with minimal needs of time for foraging)—the sleeping time is 4.0–6.5 h per night. Instead, in the wild, with a possible higher predation risk, but also with higher needs of time for foraging, the total sleeping time is only 2.1 h per night [182,183] (Table 1).

Therefore—leaving aside the insufficient data on hippopotamuses (discussed in the next paragraph)—we must conclude that the presumed inverse correlation between predation risk and sleeping time does not hold for adult megaherbivores. Indeed, saying that the reduced amounts of sleep observed in these animals is explained by an increased predation risk, is equivalent to say that elephants, rhinoceroses, and giraffes suffer the greatest predatory risk of all mammals, a fact that is obviously untrue. Instead, Table 1 shows that megaherbivores, in addition to showing highly reduced total sleeping time, spend 66% of their daily time (16 h) foraging. Therefore, the reduced sleeping time of megaherbivores is basically independent of high predatory risk but to the need of sufficient time for collecting and digesting the high amounts of forage they need.

Hippopotamuses are, in practice, predation free [174,175,176]. Therefore, they should show a large amount of total sleeping time. However, the precise figures for their sleep remain unknown. During light time, they spend 13.8 h (54.75% of total daily time) submerged in shallow pools. The submerged time is currently considered to be sleeping-resting time; but resting and sleeping are, in principle, two different states. They also sleep on land for a time that, according to Lyamin, ref. [196], occupies 10–19% of total daily time. Of course, such an ample range does not allow for a precise analysis. Regarding the foraging time, the problem is similar: on approaching darkness, hippopotamuses leave their pond and walk ~10 km [196] (up to 60 km in dry seasons). According to Lewison, they eat approximately 35–50 kg of grass daily [197]. So, we only know the amount of ingested grass but not their total sleeping time nor their total foraging time. Therefore, Table 1 only describes the averages for elephants, rhinoceroses, and giraffes.

### 7.7. Predation Risk and Diet

Apart from size, the diet is a major factor explaining why prey animals show reduced sleeping time. Mega- and meso-herbivores spend huge proportions of their time daily foraging (Table 1 and Table 2). Such figures are a consequence of the low digestibility of grass, the main food of herbivores. Cellulose is the basic component of grass and is the most abundant natural polymer on earth [92,101,187,198]. Cellulose and its related compounds are molecules with high energetic contents and therefore should show a high nutritional value. But, they are insoluble and refractory to chemical attack and can only be digested with the aid of symbiotic microorganisms cultivated in different compartments of the digestive system of some herbivores. In fact, the true food of herbivores is not grass; it is instead the microbiological flora they cultivate within their digestive system. So, herbivores provide food to the gastrointestinal flora, which in turn provide food for their hosts.

In addition to the problems of cellulose digestion, the foraging processes of herbivores implies searching for grass, biting, and masticating it, and, in many species, ruminating. Adult cattle spend 7–8 h ruminating every day, ~30% of the 24 h total daily time [199,200], with variations that depend on the quality of the available roughage. But, a significant part of rumination occurs while the animal is drowsy, i.e., in an intermediate state between wakefulness and sleep, with progressive increases in EEG amplitude (a sign of impending sleep. Ruckebusch [13] studied the drowsiness in hindgut fermenters (horses and pigs) and in ruminants (cows and sheep). In cows, drowsiness occupied 52% of the total daily time, with low arousal thresholds for auditory stimulation. However, the thresholds increased tenfold during slow-wave sleep. Therefore, it seems that many herbivores maintain acoustic vigilance not only when they are awake but also during their rumination time and are only vulnerable when they are asleep (4.12 h daily, Table 2). To summarize: ruminant meso-herbivores spend over 66% of their daily time foraging and only 17% sleeping [201,202].

Rumination is considered to be the most efficient alimentary adaptation of herbivores. However, despite being less efficient, the most frequent method for cellulose digestion is the hindgut fermentation that is observed in non-ruminant herbivores and even in omnivores, including humans [203,204,205,206,207,208]. Horses, archetypes of hindgut fermentation, possess an enlarged cecum in which—as we saw for megaherbivores—a complex microbiological flora is cultivated for cellulose digestion. However, the cecum is placed at the end of the intestine and an important part of the pre-digested food is lost in feces [206,207,209].

Nevertheless, it has been confirmed that the efficiency of hindgut fermentation is higher for rich, less fibrous foods [206,209]. Be that as it may, the total time spent in meeting the nutritional needs of both hindgut fermenters and ruminants is similar. Furthermore, it has been found that (1) the length of the digestive tube, (2) the number and dimensions of the stomach and cecal cavities, and (3) the intestinal transit time are, for all mammals, directly dependent on the concentration of cellulose in the diet [207,208,209]. In summary, the large amounts of time spent foraging and the reduced sleeping time suggest that their alimentary needs exceed, by large, those of sleep. In other words, the reductions in the sleeping time of meso-herbivores are a consequence not of predatory risk but of their foraging needs.

**Table 2 biology-14-00352-t002:** Predation risk, diet, and sleep in a sample of medium (BW: 100–700 kg) ruminants and hindgut digesters.

	BW (kg)	Sleep-Rest Time (h)	Foraging Time (h) and % Daily
African Buffalo (*Syncerus caffer*)	~650	6.63 h [210]	18.02 h (75.1%) [211]
Tsessebe (*Damaliscus lunatus*)	~120	1.92 h [212,213]	18.0 h (75%) [214]
Blue wildebeest (*Connochaetes taurinus*)	~215	4.8 h [211]	18.56 h (77.2%) [211]
Horse (*Equus caballus*)	~600	2.9 h [212]	20 h (83%) [210,211]
Donkey (*Equus asinus*)	~450	3.3 h [212]	11.18 h (46.6%) [213]
Tapir (*Tapirus Terrestris*	~235	4.4 h [212]	10.56 h (44%) [214]
Sheep (*Ovis aries*)	~100	3.8 h [212]	12.6 h (52.5% [215]
Cow (*Bos taurus*)	~285	4.0 h [212,216]	20 h (83.3%) [213,214]
Goat (*Capra aegirus hircus*)	~100	5.4 h [212]	13.8 h (57.5% [215]
Average	306 kg	4.12 h (17.1%)	15.86 h (66.08%)

#### Predation Risks and Diet

Table 1 and Table 2 show that the figures for sleeping time are almost identical for mega- and meso-herbivores (3.9 h and 4.2 h, respectively). In contrast, the predation risk is quite different: megaherbivores are almost predation-free, while the risk is extremely high for meso-herbivores that, in fact, are currently included within the preferred prey of large carnivores [114]. It is evident, therefore, that if the sleeping time depends on the predation risk, the sleeping figures for mega- and meso-herbivores would be quite different, which is uncertain. Therefore, the predation risk must be excluded from the list of factors responsible for sleeping time, for megaherbivores as well as for meso-herbivores.

From a different viewpoint, the foraging times observed that in mega- and meso-herbivores (16 h and 15.8 h, respectively) surpass 50% of the total daily time. It seems, therefore, that the need for foraging surpasses the need for sleep.

It can be argued that the values shown in Table 1 and Table 2 only represent the traits of a partial, nonrandom sample of prey. Therefore, the obtained conclusions may not be representative of the population. But the question is not whether the species listed in the tables are representative or not, but whether the rule explains the relation between sleep and predation. In fact, Table 1 and Table 2 contain many species for which the rule does not hold and, even in the case of being due to sampling errors, the exceptions represent notable reductions in the validity of the rule. Predation is unrelated to sleeping time.

### 7.8. Sleep in Predators

The following paragraphs analyze the sleep and foraging times in medium sized herbivores (Table 2) when compared with those of one of their main predators, i.e., lions.

It is possible that when remarking on the differences in the sleeping times of predators and prey, most readers think of lions predating gazelles and wolves predating lambs, perhaps disregarding the generality of the relationship between prey and predators. However, many reports have studied other predator species, obtaining similar results [217] for lynx; ref. [153] for wild dogs; ref. [218] for the spotted hyena; ref. [219] for golden jackals; ref. [220] for silver-backed jackals and golden jackals. So, we continue analyzing the case of lions. Lions rest and/or sleep between 73% and 87% of the day [221,222,223,224]. The highest values correspond to captive animals, whose hunting efforts are minimal [225,226]. Instead, lions hunt in the wild by performing short pursuits in which they expend huge amounts of energy [224]. But, in the case of success, they rapidly consume large amounts of meat [223], a food that is easily digestible. It has been found that, for a lion, it is enough to kill a prey and then eat up to 45 kg of meat every 2–4 days [227]. One may thus ask: what lions do during their nonforaging time? Simply, they perform other activities that, in fact, basically include idling, sleeping, and reproducing [227]. So, the first half of the rule shows that predators sleep for large amounts of time is true; we confirm that, in contrast with their preferred prey, meso-herbivores, time is a low-value resource for predators [228]. This is a consequence of the low risk of being predated, a fact that is true for lions. But, most likely, the large amount of sleep observed in lions is basically dependent not on the predator–prey dimension but on the large amounts of meat they eat, together with the high energy density of their food.

### 7.9. Sleep and Predation in Prey Weighing Less than 100 kg

Several studies analyzed the sleeping time of different mammalian orders, but the majority only aimed to distinguish the predation risk during NREM and REM sleep (with rather poor results). In Table 3, we list not the data of individual species but the averaged results of several animal orders. We did so to clarify the total sleeping time in the hope that reporting the average answers the question without reducing the significance.

First, we see that despite the different sources of the data presented in Table 3 and despite not showing specific data, the averages calculated for every animal order are quite comparable between studies. We thus consider that the different reports show high coherence and are indicative of the quality of the data.

The animals listed in Table 3 show multiple diets: they are herbivores, omnivores, insectivores, folivores, or carnivores [230,231,232,233]. But, they are also prey. Notably, Tapirs and Lagomorpha are exclusive herbivores and show lesser amounts of sleep compared with the rest of the group: Lagomorpha [234,235]. On the other hand, pinnipeds only show REM sleep when submerged in water and not on land [196]. Therefore, if we exclude the herbivores and marine mammals from Table 3, the remainder sleep up to 59.7% of the day (14.35 h daily). Only Pinnipeds show low amounts of sleep (4.76 h daily on land), but this reduced figure may be explained by the absence of REM sleep when they are submerged. So, it seems that the inverse relation between foraging time and predatory risk holds for the species listed in Table 3 (except for herbivores and pinnipeds). Indeed, the total sleeping time of small mammals is 14.35 h, that is, over 50% of day.

## 8. Conclusions

The present report analyzed the currently presumed hindrances caused by sleep. Many authors believe that sleep takes away time from foraging, reproductive activities, and antipredation efforts. But, we found no justification for this belief.

First, we observed that, in general, the reports describing the cause/consequence relationships between sleep, foraging, reproductive, and defensive activities are, in most cases, contradictory. Second, we showed that some of the presumed hindrances attributed to sleep are in fact benefits: if sleep reduces food intake, general health improves and lifespan increases. Likewise, if sleep reduces reproductive efforts, the results are beneficial for the biological efficiency of the species.

In fact, we showed that animals—and humans too—can reduce, and even completely suppress, sleep when hungry, when searching for reproductive partners, or when exposed to high predatory stress. Therefore, the presumed benefits that are expected for compensating for the costs of sleep are nonexistent: when other vital needs arise, all animals reduce, with no difficulty, their total sleeping time. Of course, the reduction is always limited, and the total suppression of sleep is impossible, as was demonstrated in the case of Randy Gardner: he suffered multiple mental and psychological disturbances and psychotic hallucinations during a sleep deprivation period. But surprisingly, recovery was complete after the next two days in which 14 h + 8 h of sleep was enough for total recovery [212,229]. In addition, no record exists of death simply caused by sleep deprivation. Of course, reductions in sleeping time may contribute to death, but such deaths are always the consequence of previous problems that may have been exacerbated during the period of sleep deprivation [236,237,238,239,240,241,242,243,244,245].

Nevertheless, after having discarded the idea that sleep may detract time for foraging, reproductive, and defensive activities, we reached the conclusion that the importance of sleep is low and ranks below foraging, procreating, and defensive activities. Although this is right, this is only a partial conclusion. The previous paragraphs confirmed that problems due to sleep loss appear not during the period of deprivation but during the following wakeful period. Wakeful animals must eat, procreate, and avoid predation, i.e., the most important life-sustaining activities. However, this is because sleep is a prerequisite for efficient wakefulness. Indeed, a good quota of refreshing sleep decreases the number of errors, increases working efficiency and productivity, and improves general well-being [218,219,220,221,222], but the entire set of such benefits occur not when humans or animals sleep but during the subsequent wakefulness. We observed (Section 7.3.2) that eye closure greatly increases the predatory risk in solitary drowsy animals, i.e., in those with sleep debt because of previous deprivation.

We therefore concluded that the true hindrance lies not because of sleeping but in the sleepiness that often occurs during wakefulness, a sleepiness that is always dangerous. Indeed, the number of deaths provoked by sleepiness when driving is the most important cause of death in young humans [223,224,225,226,227]. But, it is undisputable that throughout history—before the advent of cars—many deaths occurred because of sleepiness in both humans and animals. The genes promoting a sufficient quota of sleep were transmitted to their descendants that also sufficiently slept, while the genes of those that disregarded the risks of sleepiness disappeared: the risk of death is the most powerful evolutionary pressure. It is impossible to minimize the importance of this conclusion: while most hypotheses up to now tried to explain why we sleep, they only considered some advantages of sleep of relatively low and disputed entity the hypothesis presented here that considers the need for sufficient sleep as a question of life or death. We must remember that natural selection selects some individuals allowing them to survive and to kill others. And, the number of animal and human deaths caused by sleepiness during waking time represent the strongest selective force.

As an afterthought, we ask ourselves why sleep deprivation is so painful. But an easy answer exists to this question: all pleasing stimuli are pro-homeostatic and therefore increase the probability of survival. Conversely, all displeasing stimuli are anti-homeostatic and promote the risk of deep physiological disturbances that might lead to death [246,247,248]. These stimuli forced the evolutionary development of the neural circuits needed to produce a strong (but highly adaptive) displeasure to sleep deprivation.

Often, the consequences of pleasing or displeasing stimuli appear immediately, but delayed punishments—or rewards—can also modify behavior [246,247,248]. Indeed, the brain of sleep-deprived animals—or humans—says “be careful in controlling your sleep if you want to live another day”. And, as a consequence, sleep-deprived subjects make desperate efforts to overcome the deprivation and guarantee a satisfactory sleep quota; such efforts are rewarded by survival on the next day. This is the reason why animals and humans suffer the pain of sleep deprivation, and the pleasure of a sufficient sleep quota is always present in today’s humans and the animals with the capacity for hedonic experiences. Therefore, the answer to the greatest mystery of sleep is evident: “We sleep to survive the next wakeful period”.

But such an answer is still incomplete: one should ask that if sleep deprivation is so dangerous, why did animals not evolve to be capable of continuous wakefulness? As an answer, life originated on a rotating planet, with dark/light transitions every 24 h. Such cycles forced the appearance of specialists in the dark, others during light time, and others in crepuscular time. The specialists always displace the generalists [249]. Indeed, the Earth’s rotation blocks the success of generalists capable of continuous wakefulness.

## Figures and Tables

**Figure 1 biology-14-00352-f001:**
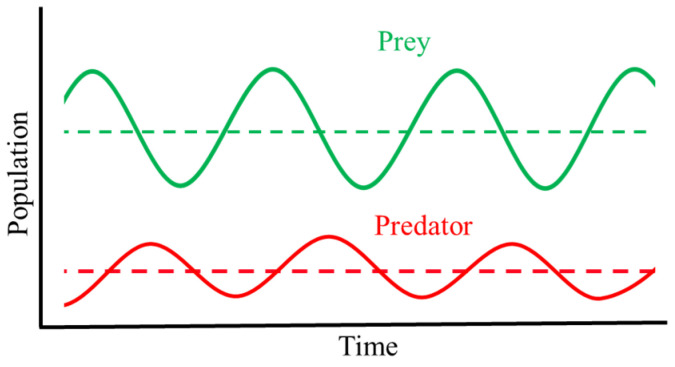
Graphic representation of the Lotka–Volterra equations in a stabilized population. The number of prey (green line) is always higher than the number of predators (red line) [102,103]. Indeed, the oscillations in the population of predators follows, with a lag, those of prey, and, in stabilized situations, the average size of the two populations remains constant, as shown by the dotted lines representing the average population of prey and predators.

**Table 1 biology-14-00352-t001:** Predation risk, foraging time, and sleep-rest time in megaherbivores (BW > 1000 kg).

Species	Predator	Predation Risk	Daily Foraging Time	Daily Sleep/Rest Time
African/Asian Elephant (*Loxodonta africana*/*Elephas maximus* 2500–6000 kg)	Lion (rare)	Invulnerable (adults) [184,185]	74.2% = 17.8 h [186]	4.0 h [182,183]
*White rhinoceros* (1000–3500) kg (*Diceros bicornis*)	Lion (rare)	Invulnerable (adults) [187]	54% = 12.86 h [188]	8 h URL (accessed on 17 April 2024) [189]: https://rhinos.org/blog/how-much-do-rhinos-sleep/
Giraffe (800–1200 kg *Giraffa camelopardalis*	Lion (rare)	Low (adults) [190,191]	54% = 12.96 h [190,192]	4.65 [193,194]
Hippopotami (*H. amphibius*) 1300–200 kg	Lion, crocodile (rare)	Very low [174,195]	>4.8 h% = 5 h [188,196]	14.04 h [188,196]
Average			16 h (excluding hippos)	16 h (excluding hippos)

**Table 3 biology-14-00352-t003:** Average sleeping time of small < 100 kg mammalian prey.

Authors: Orders:	Zepelin and Rechstchaffen (1974) [229]	Campbell and Tobler, (1984) [212]	Elgar et al. (1988) [173]	Nunn et al. (2016) [37]	Average (% of Sleeping Time)
Rodents	12.98 h	12.7 h	13.47 h	-	13.05 h (54.37%)
Insectivores	12.02 h	12.85 h		-	12.43 h (51.79%)
Non-placental	12.65 h	14.41 h	15.58 h	-	14.21 h (59.2%)
Lagomorpha	8.4 h	8.8 h	8.71 h	-	8.63 h (35.95%)
Edentates	18.3 h	13.54 h	15.34 h	-	15.72 h (65.52%)
Primates	11.47 h	10.01 h	10.32 h	11.32 h	10.78 h (44.91%)
Phalangers	13.7 h	-	-	-	13.7 h (57.08%)
Chiropters	19.7 h	-	19.8 h	-	19.75 h (82.3%)
Tapir	6.2 h	4.4 h	-	-	5.3 h (22.08%)
Pinnipeds	-	-	4.76 h	-	4.76 h (19.83%)
Marsupials	-	-	15.43 h	-	15.43 h (64.29%)
Hyracoidean	-	-	5.16 h	-	5.16 h (21.5%)
Average	12.82 h	10.95 h	12.06 h	11.32 h	12.41 h 51.7%)

## Data Availability

Not applicable.

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
