# Peer review of "The Disputable Costs of Sleeping"

_biology, 2025, doi:10.3390/biology14040352_

Round 1

Reviewer 1 Report

Comments and Suggestions for Authors

see the file enclosed

Reviewer 2 Report

Comments and Suggestions for Authors

This is an interesting manuscript, which, in general, can be useful for a wide audience. However, it may be also suitable for a book chapter.

Major comments:

1.     I think that the title should be modified within the scope of the manuscript by specification that the need in sleep was considered namely in animals. Humans may have the other needs as well such as competence, autonomy and social interactions according to Psychology.

2.    The authors should add the country name in the affiliations.

3.  I think that foraging and reproductive activity reduction caused by sleep are the clear and obvious benefits. However, L. 184-185 are not clear within it. Who rank the advantages provided by sleep low? Within the previous text the advantages of sleep reducing the risk of metabolic syndrome seem to be ranked high, so should be.

4.   L. 294-295: Are the animals with incapacity to perceive the risk abundant in the nature? I think: no.

5.  The regarded point such as the sleeping increases the risk to be predated seem doubtful. There is another meaning that sleeping allows preys to hide from predators is widely accepted. Sleeping implies low activity and movement, which let the animals not being noticed by predators. Also, the comparison of sleeping time per day in preys seem to be not sufficient. However, authors mentioned it. For example, rodents should not be forgotten. So, if the conclusion may not be valid for the whole population how can it be generalized then? I think it is a logical error. I recommend to change its interpretation and presentation.

6.      L. 407-408: According to Table 1 hippopotamuses do show larger amounts of total sleep compared to other considered megaherbivores.

7.   L. 485-486 not true due to it was observed above, not in the following paragraphs.

Minor comments:

1.       The needs to discuss electrophysiological methods in the first paragraphs seem to be low because the methods are not discussed further. L. 43-44: The sentence seems to be less informative. NREM and REM should be deciphered or their meaning should be explained when first mentioned.

2.       Paragraph 3: For me the listed statements do not seem contradictory. The option what to choose a) or b) depends on the intentions and needs of the individual. However, their consideration in such a way is also fair, however resembles demagogy.

3.       L. 252: Lotka.

4.       L. 500: kg.

Comments on the Quality of English Language

/

Reviewer 3 Report

Comments and Suggestions for Authors

I have carefully reviewed the manuscript titled “The Disputable Costs of Sleeping” by Rial et al. After a detailed evaluation, I regret to inform you that I must recommend the rejection of this manuscript for publication in Biology. I understand that addressing a complex topic like sleep and 'The Disputable Costs of Sleeping' is challenging and that many preliminary issues, especially for different taxonomic groups, are not clearly or thoroughly addressed. However, this review has several deficiencies, which I will outline below. In a review, an objective compilation of all current hypotheses is expected, accompanied by citations of the studies supporting each hypothesis. Such a summary helps to identify patterns that may explain the variability in the evidence supporting different hypotheses. However, the authors of this review did not objectively consider all the current hypotheses (see below), but also approached the subject by treating one hypothesis as true while seeking exceptions to disprove those they deem false, citing selective examples. Moreover, they disregard the concept of multicausality, analyzing each subtopic in isolation and expecting the predictions of each hypothesis to be perfectly fulfilled, without considering the covariates that should be accounted for given the multifactorial nature of the problem being addressed. There are several questionable statements throughout the manuscript, therefore I will address only a few due to its length. I hope these examples will serve as a guide that can be applied to the rest of the manuscript.

1-      The definition of sleep to be used throughout the review should be clearly outlined at the beginning of the paper. The authors initially define sleep in behavioral terms, based on eight characteristics that encompass the entire animal kingdom (see section “Sleep in animals”). They state, “In behavioral terms, sleep is currently defined as a state...” However, the references provided date back to 1913, 1973, 1981, and 1982, which does not align with the claim of a "current" definition. Additionally, the only reference from 2018 pertains exclusively to characteristic 7. In line 39 they state “most behavioral traits of sleep can also be observed in wakeful resting animals”. Does this mean that behaviors exhibiting seven out of the eight (according to lines 37-38) characteristics will be considered sleep? Or is it sufficient if only two of these characteristics are met? Can any be disregarded, or are there certain characteristics that must always be present for a behavior to be classified as sleep? This is important because in line 95 they state “…we will focus solely on sleep as a behavior”. The terms REM and NREM are introduced abruptly (lines 55-56 and 60-61), without prior explanation or definition, implying that true sleep includes these phases (there are no citations or explanation for this). How are REM and NREM related to the eight behavioral characteristics? The authors ultimately dismiss the sleep of invertebrates and reptiles simply because it differs from that of mammals and birds (Lines 45-47 and 59-60), giving the impression that they are looking for excuses to justify focusing only on mammals (and maybe birds). They also argue that it is difficult to distinguish between sleep and wakeful rest in invertebrates and reptiles, but in lines 84-86 they state “we would like to ask to the reader what we should name a paralyzed state with closed eyes that was repeated, day by day, during uncountable millions of years? Undoubtedly, such a state is sleep”. Under that definition, the behavior of reptiles must also be considered sleep. Therefore, the review should begin by clearly stating the definition of “true sleep” or sleep, either by referencing a specific definition from the literature and citing relevant sources, or by proposing a new definition. This provides sufficient justification to focus only on mammals or birds if they want to include them. The confusion regarding the definition of sleep is so substantial (see section “The birth of mammals and their sleep”) that, despite using many comparisons with birds throughout the manuscript (see below), the authors end up stating in lines 94 and 95:”Because of these doubts, the present report will deal only with the presumed costs of mammalian sleep…”.

2-      The relevant literature should be cited immediately after the argument or sentence being referenced. It is quite confusing to find entire paragraphs without references (e.g., section “Predators and preys: costs and benefits” line 235, “The concepts of Optimal Foraging and Optimal Preys” line 263, “The perception of the predatory risk by contact sensory receptors” line 297, “Vision” line 307), or all the citations grouped (e.g., Lines 138 and 213). It becomes impossible to differentiate between the authors' own arguments and published results. Additionally, it is impossible to assess the number of citations in agreement with each statement.

3-       The authors have difficulty distinguishing between different hypotheses and seeking evidence to support or refute them. For example, in lines 25 and 26, the authors stated “we show that the optimal preys are the immature, weak, sick and senescent animals and rarely the sleeping fit adults”. When we aim to understand whether sleeping has a cost compared to not sleeping, we need to compare individuals with the same characteristics (sex, age, diet, etc.) who sleep versus those who do not (this is known as a counterfactual in statistics). The fact that juveniles are the preferred prey does not invalidate the possibility that among individuals of the same developmental stage (e.g. juveniles), those who sleep might be more exposed to predation than those who do not. This logical approach should be used to assess the existing evidence in light of current hypotheses, rather than mixing different hypotheses. At no point does the hypothesis claim that individuals who sleep are more vulnerable than juveniles.

Another example of this difficulty appears in lines 132-134. There is no ambiguity in the definition of the causes and consequences of sleep; rather, there are different hypotheses, each with varying causal and consequential relationships. One hypothesis may likely be valid for organisms of a certain size and/or with specific survival strategies or context, and another hypothesis may apply to organisms of a different size and/or with other survival strategies or contexts.

4-      The authors compare results, without distinguishing between different contexts. For example, they compare the behavior of migratory birds (although they said they focus only on mammals), which fly all night and must sleep and eat during the day (as in citation 28, for example), with pathologic conditions such as metabolic syndrome in humans, to argue that the reduction in foraging time due to sleep is not a cost but a benefit because it improves general health and extends lifespan. The pathological opposite of metabolic syndrome, according to this line of reasoning (which I consider biased and lacking scientific basis), would be anorexia. To assess the cost of sleeping on foraging time in migratory birds, the appropriate counterfactual would be those same birds during the non-migratory season. Does the time spent eating and sleeping vary between migratory and non-migratory contexts? Additionally, rather than assuming it is beneficial because it extends lifespan, it should be evaluated how the reduction in foraging time impacts the ability to fly and arrive safely at the destination (among many other factors). In a review, it is expected to see references to studies that have properly assessed these costs using a sound experimental design and statistical analyses and to compare such evidence across different species and animal groups. For this purpose, it is not feasible to conclude from random situations and raw datasets from the literature. Contexts are factors, covariates, that must inevitably be considered when debating the evidence for and against a hypothesis.

A similar situation appears in lines 360-369 when the authors discuss Scotophobia and agoraphobia.

5-      In the section titled "Sleep and Reproductive Efficiency" (line 186), the authors state “Many authors affirm that sleep detracts time from reproductive activity” without references. They again abruptly introduce new terms such as “reproductive efficiency” and “biological efficiency” (although these terms are defined later, it is unclear whether these are their own definitions or they forgot to cite relevant literature). All the paragraph is based on a critique of a single study by Lesku et al. (2012) on one species of bird (although they said they focus only on mammals), the polygynous Arctic-breeding shorebird Calidris melanotos. For reasons they do not specify, they argue that "biological efficiency," which depends on the number of fertile descendants, is what truly matters. However, to study “biological efficiency” based on their definition is quite challenging considering the length of mammalian gestation time and reproductive age. This would require conducting breeding experiments, performing paternity tests on the F1 generation, waiting for the F1 generation to reach sexual maturity, re-breeding the entire F1 generation, and assessing the number of individuals producing offspring. Furthermore, it is not clear why this invalidates the relationship between time spent sleeping and the number of descendants. In line 543 authors state “Likewise, if sleep reduces the reproductive efforts, the results are beneficial for the Biological Efficiency species”. I also don't understand how a reduction in reproductive effort affects the fertility of the F1 generation, either positively or negatively.

6-      In line 382, the relationship between body size and predation risk is discussed without addressing the connection between sleep duration, body size, and metabolic rate. It has generally been observed that in mammals, there is a correlation between body size and sleep amount, with larger animals typically sleeping less than smaller ones. This trend may be related to differences in metabolic needs as well as other survival strategies among species of varying sizes. However, while a general trend may exist between body size and sleep duration, various ecological, behavioral, and physiological factors can lead to deviations from this expected pattern. These deviations from the general body size-sleep duration relationship appear as intercept changes when covariates are included in a linear regression. This means that, under each state of the covariate, the correlation between body size and sleep duration may be recovered, but if we do not account for the covariate, we obtain a line with considerable variation around it and a low R² value. It is also necessary to consider that the relationship might be nonlinear, so we would not expect a straight line. None of this can be concluded by merely observing raw data in a table, as the authors of this review seem to propose. Moreover, exceptions to the rule do not invalidate it; they simply limit its applicability to specific situations or contexts.

7-      The manuscript lacks organization and contains inconsistencies in tables (Table 3 does not include column names), and the bibliography is not cited in order (see lines 53, 221,380,386, 407, etc) compromising its overall clarity and quality.

8-      Throughout the manuscript, the evolutionary process is referred to in a teleological manner, suggesting that nature has an inherent directionality or purpose. However, evolution occurs as a result of the interaction of natural processes such as natural selection, mutation, migration, and genetic drift, without any specific purpose. Therefore, I suggest avoiding phrases such as " These facts forced the evolutionary need for developing” (line 590), “So primitive mammals were forced to be strictly nocturnal” (line 73-74), “were forced to share their environment with terrible beasts” (lines 77-78). Additionally, referring to dinosaurs as 'terrible beasts' in this statement is both subjective and inaccurate, as it conveys a dramatic tone and implies a negative or threatening quality that does not align with the scientific understanding of these animals.

I hope my comments will assist the authors in improving the overall quality of the manuscript, and also help ensure that issues that seem poorly understood are not taken for granted or prematurely resolved based on arbitrary stances.

Round 2

Reviewer 1 Report

Comments and Suggestions for Authors

now it's o.k.